# EPOS4: New theoretical concepts for modeling proton-proton and ion-ion scattering at very high energies

**Klaus Werner[1]**

**1** SUBATECH, Nantes University – IN2P3/CNRS – IMT Atlantique, Nantes, France

*22nd International Symposium on Very High Energy Cosmic Ray Interactions (ISVHECRI 2024) Puerto Vallarta, Mexico, 8-12 July 2024*

## Abstract

**I explain the new concepts underpinning EPOS4, a novel theoretical framework designed to model hadronic interactions at ultrarelativistic energies. This approach eventually reconciles the parallel multiple scattering scenario (needed in connection with collective effects) and factorization (being the conventional method for high-energy scattering).**

## 1   Introduction

In Fig. 1, one can see the typical space-time representation of high-energy hadronic scatterings. The process begins with primary interactions occurring within a pointlike overlap zone (depicted as a red point) in proton-proton ($pp$) collisions, as well as in proton-nucleus ($pA$) or nucleus-nucleus ($AA$) collisions. Subsequently, the formation of the quark-gluon plasma (QGP) and the production of final state hadrons occur at a later stage.

In the diagram, it is evident that a comprehensive representation of space-time must consider the prior splitting of partons (parton evolution). This process (which prepares the actual

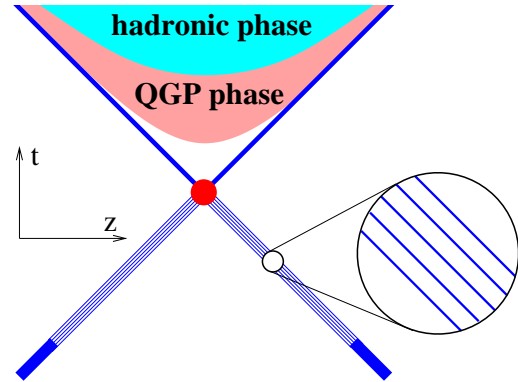

Figure 1: Space-time picture of hadronic scatterings.

| | meaning |
|---|---|
| $\mathbf{T}(s,t)$ | elastic scattering T-matrix;<br>$s$, $t$ Mandelstam variables |
| $T(s,b)$ | Fourier transformation of $\mathbf{T}(s,t)$ with respect to the momentum transfer, divided by $2s$ (impact parameter representation) |
| $G$ | $2\,\mathrm{Im}\,T$ – representing inelastic scattering (cut diagram) |
| $\tilde{\sigma}$ | Integrand in cross section formulas:<br>$pp$: $\quad\sigma^{pp} = \int d^2b\,\tilde{\sigma}^{pp}(s,b)$<br>$A+B$: $\quad\sigma^{AB} = \int db_{AB}\,\tilde{\sigma}^{AB}(s,b,\{b_i^A\},\{b_i^B\})$<br>with<br>$\int db_{AB} = \int d^2b \int \prod_{i=1}^{A} d^2b_i^A\,T_A(b_i^A)$<br>$\qquad\qquad \int \prod_{j=1}^{B} d^2b_j^B\,T_B(b_j^B)$ |

Table 1: Important symbols.

scattering) takes a long time due to significant $\gamma$ factors. However, the interaction region (depicted in red) is indeed pointlike, necessitating multiple scatterings to occur simultaneously (the long "preparation" does not allow to have the scatterings one after the other). In the EPOS4 approach for primary interactions, one avoids sequential scatterings for both parton-parton and nucleon-nucleon interactions by rigorously conducting multiple scatterings in parallel. This is true for both, the theoretical formalism and the Monte Carlo realization, based on the principle that the Monte Carlo must be derived directly from theory, which is a non-trivial task.

The EPOS4 method has been previously introduced in a series of technical papers [1–4] that span 160 pages and aim to provide comprehensive details to avoid treating EPOS4 as a black box. These papers address numerous solved technical challenges, such as $N$-dimensional integrals and probability laws with $N > 10^6$. However, beyond these technical aspects lie new and distinctive concepts that need to be clearly elucidated, along with explanations of their functionality. This is the main objective of this communication. These concepts establish a connection between pre-QCD multiple scattering approaches [5–8] and the standard tool in high-energy scattering, the factorization approach [9, 10].

Some technical remarks: I use the symbols $\mathbf{T}$, $T$, $G$, and $\tilde{\sigma}$ as explained in Tab. 1, where I use transverse nucleon coordinates $b_i^A$ and $b_j^B$, and the nuclear thickness function $T_A(b) = \int dz\,\rho_A\big(\sqrt{b^2+z^2}\big)$, where $\rho_A$ is the (normalized) nuclear density for the nucleus $A$.

Let us start by delving into the Gribov-Regge (GR) approach, as documented in [5–8]. In

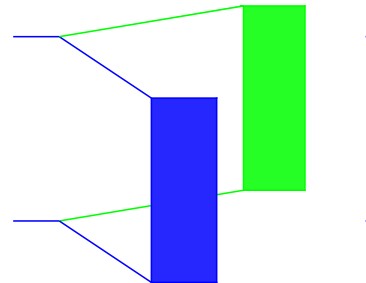

Figure 2: Double scattering in GR.

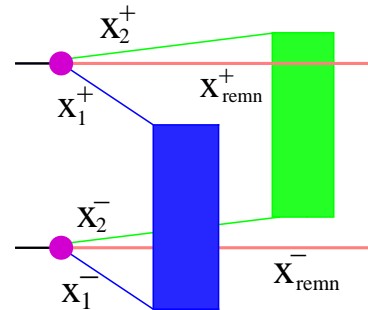

Figure 3: Energy-momentum sharing (GR$^+$) in EPOS4.

this approach, multiple scattering in $pp$ occurs in a strictly parallel manner, as illustrated in Fig. 2. Each box represents an inelastic subscattering $G$, leading to chains of particles, with the specific mechanism being unknown at that time. There is no conflict even when a lengthy "preparation" is required, as discussed earlier. In the GR approach, cross sections are expressed in terms of weights $P$ that depend on the single scattering expression $G$ as

$$\tilde{\sigma}_{\text{in}}^{pp} = \sum_{m=1}^{\infty} \underbrace{\frac{1}{m!} G^m \, e^{-G}}_{P(m)}, \quad \tilde{\sigma}_{\text{in}}^{AB} = \sum_{\{m_k\}} \underbrace{\prod_{k=1}^{AB} \frac{1}{\text{m}_\text{k}!} (G_k)^{m_k} \, e^{-G_k}}_{P(\{m_k\})} \tag{1}$$

where the terms $P(m)$ and $P(\{m_k\})$, representing probability distributions, may serve as a basis for Monte Carlo applications. Let me discuss, step by step, the improvements realized in the EPOS4 approach.

## 2   Adding energy-momentum conservation

In some cases, energy-momentum conservation is not particularly significant (such as for total cross sections), but for other cases, it is absolutely essential (like in particle production). It is also necessary as a solid basis for Monte Carlo applications. As discussed in detail in Ref. [3]: To ensure energy-momentum sharing (GR$^+$) in EPOS4, in $pp$ or for each $NN$ scattering in $A+B$, one considers (compared to GR) new variables: the lightcone momentum fractions $x_m^+$ and $x_m^-$ of subscatterings, with

$$x_{\text{remn}}^\pm = 1 - \sum x_m^\pm \,, \tag{2}$$

being the lightcone momentum fraction of the remnant, see Fig. 3.

The expressions for cross sections, as shown in Ref. [3], still use weights $P(K)$ for configurations

$$K = \left\{ \{m_k\}, \{x_{k\mu}^\pm\} \right\}, \tag{3}$$

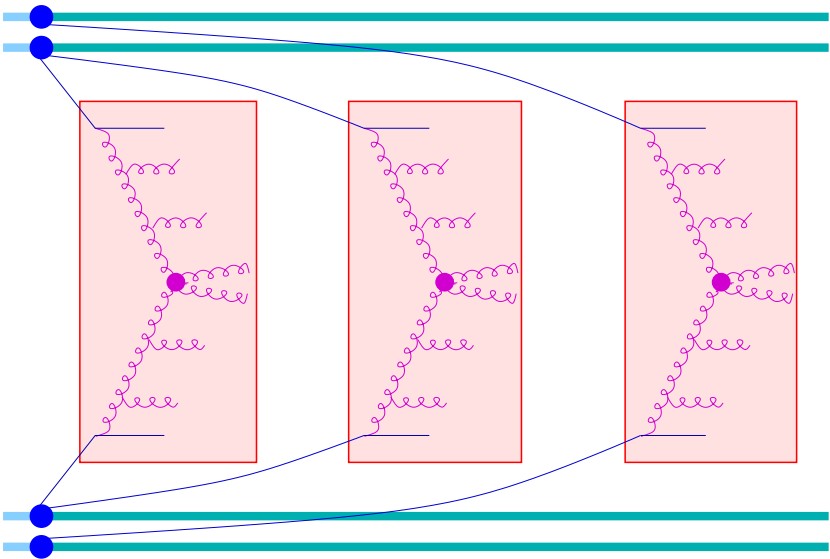

Figure 4: Using $G = G_{\text{QCD}}$.

referring to $m_k$ subscatterings per pair $k$, with lightcone momentum fractions $x^{\pm}_{k\mu}$.

This provides a solid basis for Monte Carlo simulations: one determines $K$ according to $P(K)$, instantaneously, there are no sequences, everything happens in parallel. And one has MC = theory.

## 3   Making the link with QCD

The current framework is founded on "some $G$" where $G$ denotes a subscattering. The next step involves establishing the connection with QCD. It is assumed that

$$G = G_{\text{QCD}} , \tag{4}$$

with $G_{\text{QCD}}$ denoting parton-parton scattering based on pQCD, incorporating DGLAP evolution. (see Ref. [2] and early work (no heavy flavor) in Ref. [11]). This means one replaces the boxes of the GR approach with QCD diagrams, as sketched in Fig. 4 for a collision of two nuclei with three subscatterings.

One calculates and tabulates "modules" (QCD evolution, Born cross sections, vertices), which enables one to assess the diagram. Various methods exist for reorganizing the modules, and one option is to establish (and tabulate) a parton distribution function (PDF), enabling the computation of the jet cross section versus $p_t$ for pp at 13 TeV (see Ref. [2]) as shown in Fig. 5. The red line represents the EPOS result, and it is compared to ATLAS data [12] (triangles) as well as results derived from CTEQ PDFs [13] (green dashed line). It seems that everything is under control, but here one considered just one single subscattering.

In the Gribov-Regge approach, the full multiple scattering scenario is (up to a factor $AB$) equal to the single one for inclusive cross sections (AGK theorem), i.e.,

$$\frac{d\sigma^{AB}_{\text{incl}}}{dp_t} \bigg/ AB \times \frac{d\sigma^{\text{single scattering}}_{\text{incl}}}{dp_t} \tag{5}$$

is unity. Unfortunately, as shown in [3], one gets at high $p_t$ for this ratio 0.2 and 0.5 for minimum bias PbPb at 5.02 TeV and $pp$ at 5.02 TeV, respectively. When trying to understand

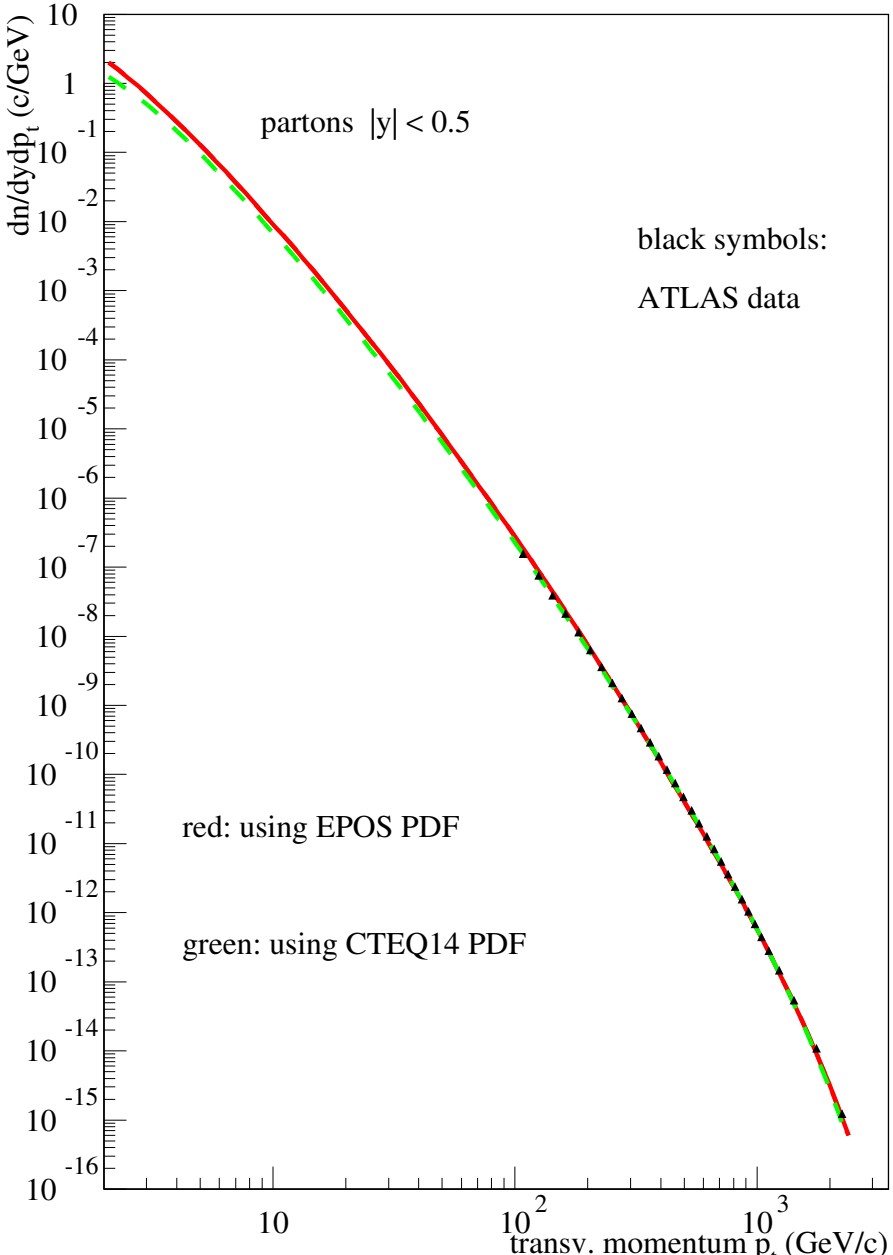

Figure 5: Jet cross section versus $p_t$ for pp at 13 TeV. The red line represents the EPOS result, compared to ATLAS data [12] (triangles) as well as to results derived from CTEQ PDFs [13] (green dashed line). The EPOS4 curve is based on EPOS4 parton distribution functions.

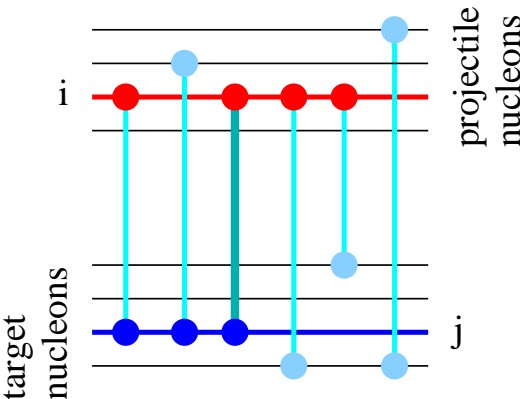

Figure 6: For a given subscattering (Pomeron), connected to projectile nucleon $i$ and target nucleon $j$, one defines the connection number $N_{\text{conn}} = \frac{N_P + N_T}{2}$ where $N_P$ is the number of scatterings involving nucleon $i$, and $N_T$ the number of scatterings involving nucleon $j$.

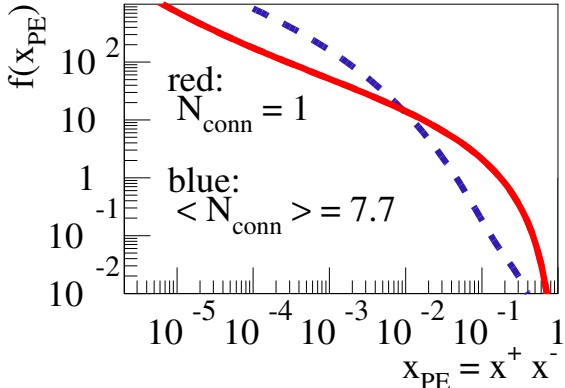

Figure 7: The $x_{\text{PE}}$ distribution $f(x_{\text{PE}})$. The red curve refers to $N_{\text{conn}} = 1$ (an isolated Pomeron), whereas the blue dashed one refers to central collisions with an average $N_{\text{conn}} = 1$ of around 7.7. Large $N_{\text{conn}}$ amounts unavoidably to large $x_{\text{PE}}$ being suppressed, due to energy-momentum-sharing.

the origin of this failure, one finds immediately (see Ref. [3]) that it is related to energy-momentum sharing among subscatterings. Inclusive particle spectra (like $p_t$ distributions) are determined by the distribution of the LC momenta $x^+$ and $x^-$ of the subscatterings. The squared CMS energy fraction

$$x_{\text{PE}} = x^+ x^- \approx s / s_{\text{tot}} \qquad (6)$$

is a crucial element, and I will explain below how the distribution of the variable $x_{\text{PE}}$ is impacted by energy-momentum sharing.

For a given subscattering (Pomeron), involving projectile nucleon $i$ and target nucleon $j$, one defines the connection number

$$N_{\text{conn}} = \frac{N_P + N_T}{2} \, , \qquad (7)$$

where $N_P$ is the number of scatterings involving nucleon $i$, and $N_T$ the number of scatterings involving nucleon $j$, see Fig. 6.

The $x_{\text{PE}}$ distributions $f(x_{\text{PE}})$ depend on $N_{\text{conn}}$. Large $N_{\text{conn}}$ amounts unavoidably to large $x_{\text{PE}}$ being suppressed, whereas small $x_{\text{PE}}$ is enhanced, as shown in Fig. 7. I will use the notation $f^{(N_{\text{conn}})}(x_{\text{PE}})$.

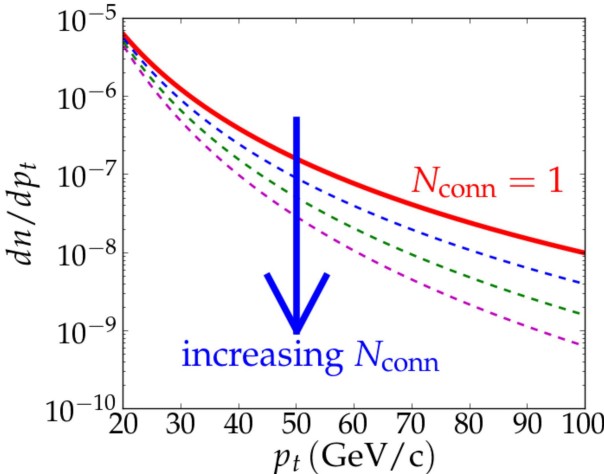

Figure 8: The suppression of large values of $x_{\mathrm{PE}}$ as a consequence of large $N_{\mathrm{conn}}$ implies a suppression of large $p_t$.

The value of $x_{\mathrm{PE}}$ is strongly correlated with the transverse momentum $p_t$ of produced particles. Pomerons with large $x_{\mathrm{PE}}$ produce with a higher probability high $p_t$ particles compared to Pomerons with small $x_{\mathrm{PE}}$. For very small $x_{\mathrm{PE}}$, the hard scattering even disappears completely, and soft Pomerons take over, to produce low $p_t$ hadrons.

Therefore, a suppression of large values of $x_{\mathrm{PE}}$ (as a consequence of large $N_{\mathrm{conn}}$) implies a suppression of large $p_t$, as sketched in Fig. 8. This is in particular true for the large $N_{\mathrm{conn}}$ contributions in minimum bias $pp$ or $AA$ scattering. So the superposition of the different contributions (of different values of $N_{\mathrm{conn}}$) cannot be equal to the single-scattering case ($N_{\mathrm{conn}} = 1$), one gets always a suppression at large $p_t$ (and therefore a violation of AGK).

As a first step towards a solution, the problem will be "quantified". One defines the "deformation" of $f^{(N_{\mathrm{conn}})}(x_{\mathrm{PE}})$ relative to the reference $f^{(1)}(x_{\mathrm{PE}})$ as

$$R_{\mathrm{deform}} = \frac{f^{(N_{\mathrm{conn}})}(x_{\mathrm{PE}})}{f^{(1)}(x_{\mathrm{PE}})}. \tag{8}$$

It is $R_{\mathrm{deform}} \neq 1$ which creates the problem. But one is able to parameterize $R_{\mathrm{deform}}$ and tabulate it, for all systems, all centrality classes (see Ref. [3]). So

$$R_{\mathrm{deform}} = R_{\mathrm{deform}}(N_{\mathrm{conn}}, x_{\mathrm{PE}}) \tag{9}$$

can be considered to be known, it is tabulated and available via interpolation (to be used later), see Fig. 9, where the red line corresponds to a simulation and the dotted one to a parameterization.

This "parameterization of the problem" will be the key element of the solution, to be sketched in the following, and discussed in detail in Ref. [3].

## 4   Adding saturation

The single scattering expression $G$, which is the basic component of the multiple scattering formalism, actually presents two issues: (i) The assumption that $G = G_{\mathrm{QCD}}$ appears to be incorrect (AGK problem), and (ii) there is a complete absence of nonlinear effects, as shown in Fig. 10 (a).

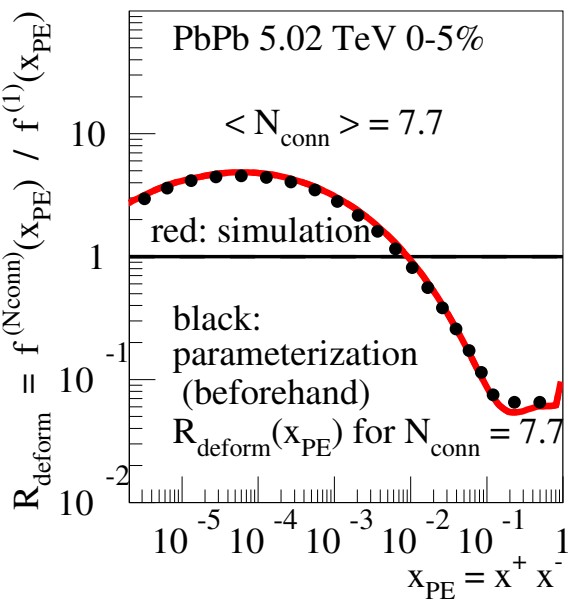

Figure 9: The deformation function, representing the change of $f^{(N_{\mathrm{conn}})}(x_{\mathrm{PE}})$ relative to the reference $f^{(1)}(x_{\mathrm{PE}})$. The red line corresponds to a simulation and the dotted one to a parameterization.

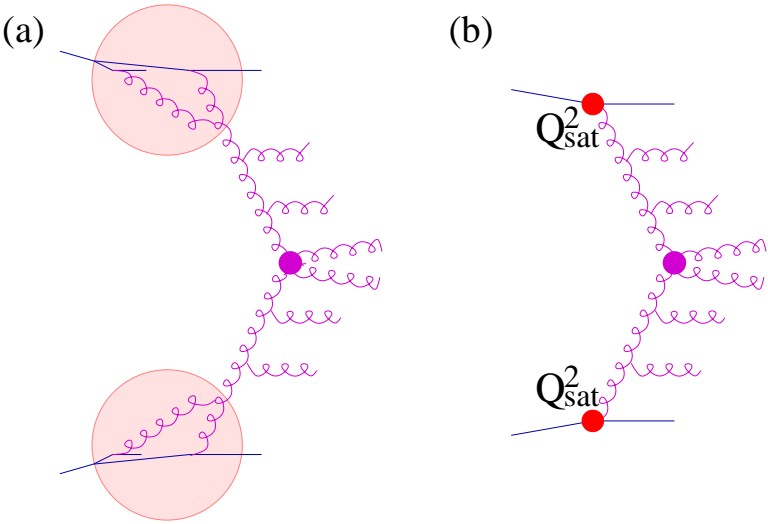

Figure 10: (a) Nonlinear effects, like gluon fusion (inside the red circles) is absent for the moment. (b) Adding nonlinear effects by introducing saturation scales $Q^2_{\mathrm{sat}}$ which are meant to "summarize" these nonlinear effects.

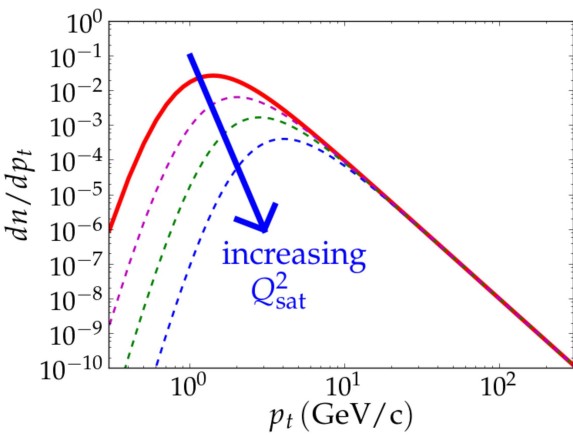

Figure 11: Contributions with increasing $N_{\text{conn}}$.

The third EPOS4 improvement amounts to adding saturation, by assuming that the nonlinear effects, inside the circles in Fig. 10 (a), may be "summarized" by saturation scales, suggesting to treat nonlinear effects by introducing saturation scales $Q_{\text{sat}}^2$ as the lower limits $Q_0^2$ of the virtualities for DGLAP evolutions, see Fig. 10 (b). One computes and tabulates $G_{\text{QCD}}(Q_0^2, x^+, x^-, s, b)$ for a large range of $Q_0^2$ values, see Ref. [2].

Concerning the connection between the basic multiple scattering building block $G$ and the QCD expression $G_{\text{QCD}}$, one postulates that for each subscattering, for given $x^{\pm}, s, b$, and $N_{\text{conn}}$, one has

$$G(x^+, x^-, s, b) = n \frac{G_{\text{QCD}}(Q_{\text{sat}}^2, x^+, x^-, s, b)}{R_{\text{deform}}(N_{\text{conn}}, x_{\text{PE}})}, \tag{10}$$

**such that $G$ does not depend on $N_{\text{conn}}$,** whereas $Q_{\text{sat}}^2$ does so. Here, $n$ is a normalization constant. Using Eq. (10), one can show [3]:

$$\frac{d^2\sigma_{\text{incl}}^{AB(N_{\text{conn}})}}{dx^+ dx^-} \propto \frac{d\sigma_{\text{incl}}^{\text{single scattering}}}{dx^+ dx^-} \left[ Q_{\text{sat}}^2(N_{\text{conn}}, x^+, x^-) \right], \tag{11}$$

i.e., the A+B cross section (for given given $N_{\text{conn}}$) is equal to the single scattering case, but with $Q_{\text{sat}}^2$ corresponding to $N_{\text{conn}}$. The same relation holds for $p_t$ distributions (deduced from $x^+x^-$). One expects, as sketched in Fig. 11, with increasing $N_{\text{conn}}$ an increasing $Q_{\text{sat}}^2$, and a reduction at $p_t^2 < Q_{\text{sat}}^2$ compared to $N_{\text{conn}} = 1$ (red curve). But no change for large $p_t$.

If one is interested in large $p_t$, one replaces $Q_{\text{sat}}^2$ by some constant $Q_0^2 = \max\{Q_{\text{sat}}^2\}$, and one gets finally

$$\frac{d\sigma_{\text{incl}}^{AB(mb)}}{dp_t} = AB \frac{d\sigma_{\text{incl}}^{\text{single scattering}}}{dp_t} \left[ Q_0^2 \right], \tag{12}$$

but only for $p_t^2$ bigger than the relevant $Q_{\text{sat}}^2$ values (a kind of generalized AGK theorem). This is extremely important: one gets (for the first time) factorization (in $pp$ and $A+B$) for inclusive cross sections at high $p_t$ in a fully selfconsistent[1] multiple (parallel) scattering scheme. What this means, is shown in Fig. 12, where the jet cross section for $pp$ at 13 TeV is plotted. This is the same plot as shown earlier, but here I add in addition the full Monte Carlo result (blue points). Important: The Monte Carlo curve agrees at large $p_t$ with the red curve, representing the single Pomeron result based on PDFs. **Without the requirement formulated in Eq. (10), the Monte Carlo result (blue) would be a factor 5 below the single Pomeron case (red).**

---

[1]Mandatory: (A) energy-momentum conservation, (B) parallel scattering, (C) MC = theory, (D) factorization for high $p_t$

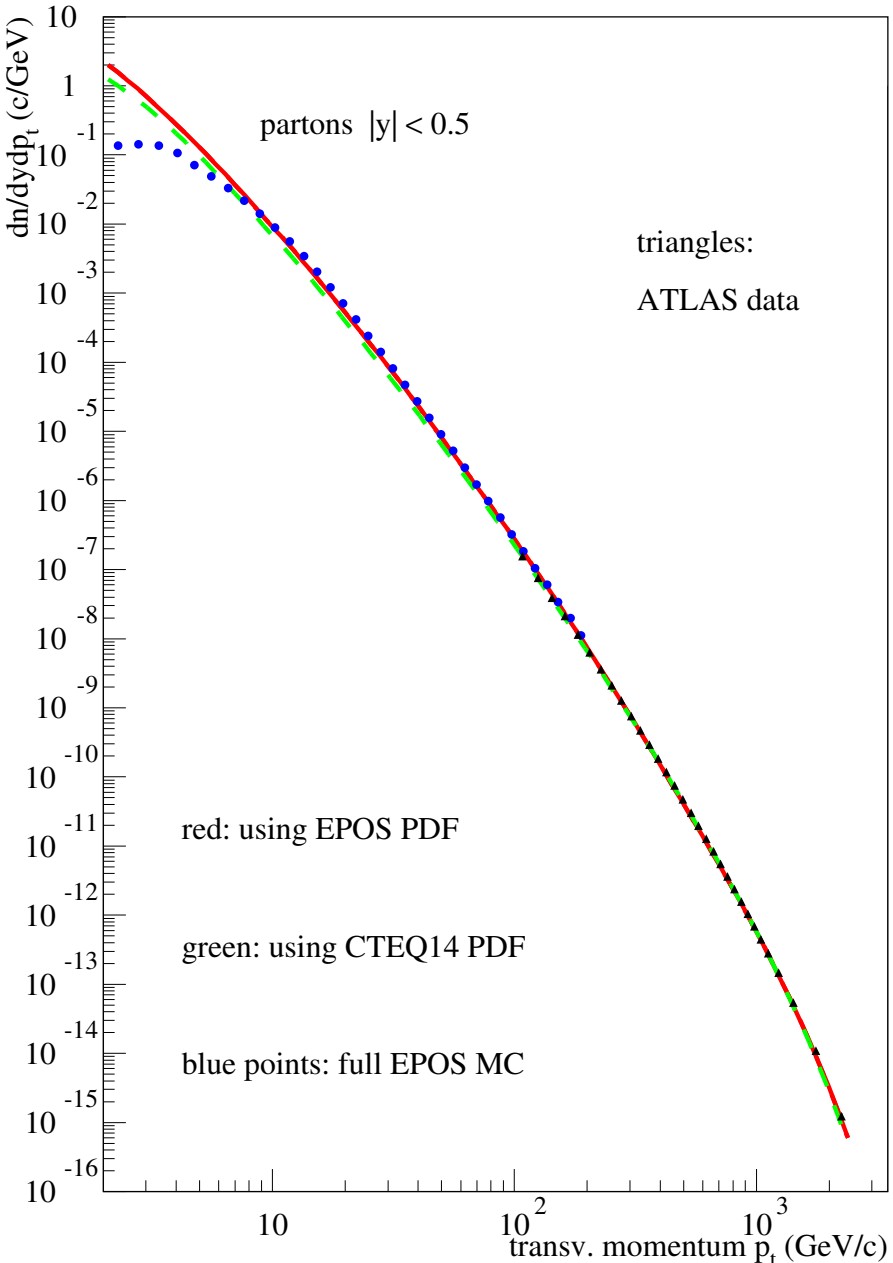

Figure 12: Jet cross section versus $p_t$ for pp at 13 TeV. The full (multiple scattering) Monte Carlo result (blue points) is compared to the EPOS result for one single Pomeron (red line) and to ATLAS data [12] (triangles). I show as well a result based on factorization, derived from CTEQ PDFs [13] (green dashed line).

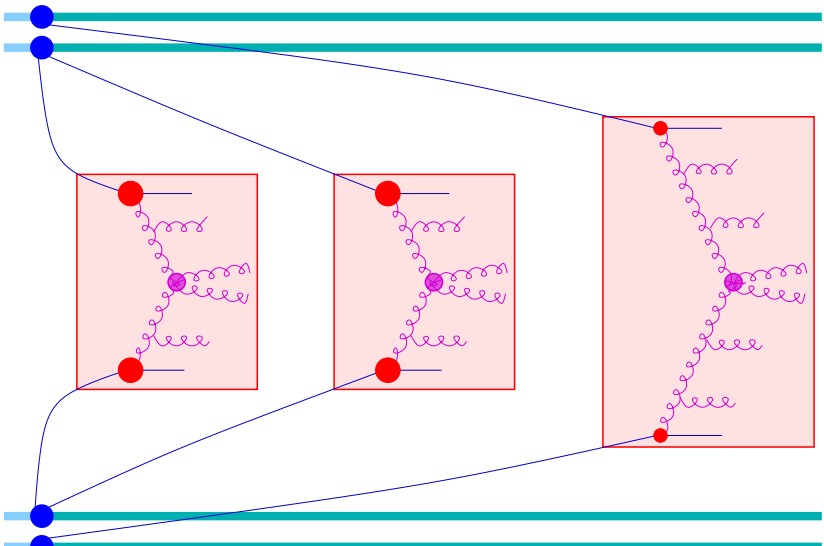

Figure 13: Sketch of the "compensation" of smaller energies (red box sizes) by larger saturation scale values (red dots).

How to understand why these $N_{conn}$-dependent saturation scales "work"? Let me qualitatively explain this by considering an $A + B$ scattering ($A = B = 2$) with 3 subscatterings, as shown in Fig. 13 (more quantitative discussions can be found in Refs. [2,3]). Let me consider any of the two left scatterings, compared to the right one: $N_{conn}$ is bigger (2 compared to 1); the energy ($\sqrt{s}$) smaller due to energy sharing; $Q_{sat}^2$ is bigger because of the larger $N_{conn}$ (bigger dots); the parton evolution shorter due to the bigger $Q_{sat}^2$; **the central part responsible for the hard scattering is identical.** This last point is the crucial element, which assures that at the end the hard particle production is identical independent of $N_{conn}$, and therefore the sum of all $N_{conn}$ contributions is (up to a factor) identical to the single Pomeron case. And this is what is needed to get factorization in such a multiple scattering formalism.

## 5   Conclusion

I explained the new concepts underpinning EPOS4. Starting from the GR approach (which ensures already parallel scatterings), I explain the improvements in three steps: (a) adding energy-momentum conservation; (b) making the link with QCD; (c) adding saturation. The latter is done such that the (unavoidable) deformation of the Pomerons energy distribution in the case of many parallel scatterings, is completely "absorbed" into the saturation scale, which ensures that at the end the high $p_t$ values are not affected. This is nothing less than a reconciliation of multiple scattering, featuring parallel scatterings, and factorization.

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
