# Peer review of "EPOS4: New theoretical concepts for modeling proton-proton and ion-ion scattering at very high energies"

_SciPost Physics Proceedings_

## Round 3 · Referee Report · Anonymous (Referee 1) · 2025-1-2

Report

The manuscript can be accepted for publication.

Recommendation

Publish (easily meets expectations and criteria for this Journal; among top 50%)

---

## Round 3 · List of Changes

1) In the Introduction, the author rightfully argues that the initial stage of high energy collisions proceeds on much smaller time-distance scales than final state interactions, which allows one to treat the two stages separately. However, in lines 17-22, he states that while the initial stage takes a long time, the corresponding interaction region is pointlike. This may confuse a potential reader. Generally, the size of the interaction region is defined by the interaction time. What the author probably means is that while the time scale of the initial stage is long enough (and the size of the respective region is large enough) to allow for multiple scattering processes to proceed in parallel, this time is very short, compared to the characteristic time scales of final state interactions. Hence, the interaction region can be considered pointlike, compared to the one relevant for final state interactions.

No I don't mean that the time scale of the initial stage is long enough to allow for multiple scattering processes to proceed in parallel. The actual multiple scattering processes proceed in a very short time interval (considered infinitely small). What takes time is the preparation, which at the end provides many partons. The long "preparation" does not allow to have the scatterings one after the other. I changed the paragraph to make this more clear:

In the diagram, it is evident that a comprehensive representation of space-time must consider the prior splitting of partons (parton evolution). This process (which prepares the actual scattering) takes a long time due to significant $\gamma$ factors. However, the interaction region (depicted in red) is indeed pointlike, necessitating multiple scatterings to occur simultaneously (the long "preparation" does not allow to have the scatterings one after the other). In the EPOS4 approach for primary interactions, one avoids sequential scatterings for both parton-parton and nucleon-nucleon interactions by rigorously conducting multiple scatterings in parallel. This is true for both, the theoretical formalism and the Monte Carlo realization, based on the principle that the Monte Carlo must be derived directly from theory, which is a non-trivial task.

2) Ref. [7] corresponds to the perturbative treatment ('GL' of 'DGLAP') rather than to the Gribov-Regge approach.

I cannot localize the problem. Yes [7] V. N. Gribov and L. N. Lipatov, , Sov. J. Nucl. Phys. 15, 438 (1972) refers to Gribov stuff. I think I use [7] to refer to this

---

## Editorial Decision

accepted_in_target_journal